# Characterization of Procoagulant COAT Platelets in Patients with Glanzmann Thrombasthenia

**DOI:** 10.3390/ijms21249515

**Published:** 2020-12-14

**Authors:** Alessandro Aliotta, Manuel Krüsi, Debora Bertaggia Calderara, Maxime G. Zermatten, Francisco J. Gomez, Ana P. Batista Mesquita Sauvage, Lorenzo Alberio

**Affiliations:** Division of Hematology and Central Hematology Laboratory, Lausanne University Hospital (CHUV) and University of Lausanne (UNIL), Rue du Bugnon 46, CH-1011 Lausanne, Switzerland; Alessandro.Aliotta@chuv.ch (A.A.); manuel.krusi@unil.ch (M.K.); Debora.Bertaggia-Calderara@chuv.ch (D.B.C.); Maxime.Zermatten@chuv.ch (M.G.Z.); Francisco-Javier.Gomez@chuv.ch (F.J.G.); Ana-Patricia.Batista-Mesquita@chuv.ch (A.P.B.M.S.)

**Keywords:** procoagulant platelets, Glanzmann thrombasthenia, procoagulant activity, ion fluxes

## Abstract

Patients affected by the rare Glanzmann thrombasthenia (GT) suffer from defective or low levels of the platelet-associated glycoprotein (GP) IIb/IIIa, which acts as a fibrinogen receptor, and have therefore an impaired ability to aggregate platelets. Because the procoagulant activity is a dichotomous facet of platelet activation, diverging from the aggregation endpoint, we were interested in characterizing the ability to generate procoagulant platelets in GT patients. Therefore, we investigated, by flow cytometry analysis, platelet functions in three GT patients as well as their ability to generate procoagulant collagen-and-thrombin (COAT) platelets upon combined activation with convulxin-plus-thrombin. In addition, we further characterized intracellular ion fluxes during the procoagulant response, using specific probes to monitor by flow cytometry kinetics of cytosolic calcium, sodium, and potassium ion fluxes. GT patients generated higher percentages of procoagulant COAT platelets compared to healthy donors. Moreover, they were able to mobilize higher levels of cytosolic calcium following convulxin-plus-thrombin activation, which is congruent with the greater procoagulant activity. Further investigations will dissect the role of GPIIb/IIIa outside-in signalling possibly implicated in the regulation of platelet procoagulant activity.

## 1. Introduction

In order to maintain physical integrity during episodes of injury and disease, primary and secondary hemostasis are important processes to limit life-threatening bleeding [1]. The first process encompasses mainly vessel wall reactivity and aggregation of platelets. The second consists of soluble clotting factors mainly synthesized in the liver parenchyma [2]. Activation of these clotting factors leads to a common endpoint: the transformation of fibrinogen in fibrin. The latter will engulf and stabilize a preformed primary clot, mainly made up of aggregated platelets, and stop the bleed.

Aggregation of platelets is mediated by the surface receptor glycoprotein (GP) IIb/IIIa (integrin αIIbβ3). In the first steps, agonists activate GPIIb/IIIa (called inside-out signaling). Activation of GPIIb/IIIa is characterized by a conformational change to increase its affinity principally for fibrinogen but also for other adhesive molecules such as fibronectin or von Willebrand factor [3]. The binding of fibrinogen to activated GPIIb/IIIa receptors allows platelet bridging and also mediates outside-in signaling to culminate with a stable and irreversible aggregation of platelets [2,3,4].

Glanzmann thrombasthenia (GT) is a rare autosomal inherited disorder, characterized by a quantitative or qualitative defect in GPIIb/IIIa, thus impairing platelet aggregation and normal primary hemostasis [2,4]. Pathogenic mutations in the genes *ITGA2B* and *ITGB3* prevent the normal function of GPIIb/IIIa receptor, weakening platelet aggregation and leading to unstable clot formation and thus to bleeding phenotype [5,6,7]. Affected patients can range in their symptoms from being nearly asymptomatic to bleeding episodes that can vary in intensity and frequency and are characterized by easy mucocutaneous bleeding and bruising [8].

In recent decades, it became increasingly clear, that platelets do not represent a homogenous population of cells but rather a heterogeneous assortment of subpopulations that differ upon activation in their structural features as well as their functional properties [9,10,11].

Platelets show great variability in their agonist-induced response patterns and the propensity to expose phosphatidylserine (PS) on their surface, which in turn is one of the hallmarks of the procoagulant platelet population [12,13].

Platelet procoagulant activity, as an additional activation endpoint to traditional secretion and aggregation, is generated upon strong platelet activation [14,15]. In particular, the combination of potent agonists such as thrombin (THR) and collagen [or convulxin (CVX), a selective agonist of the GPVI collagen receptor] induce the formation of procoagulant platelets that become highly efficient in sustaining thrombin generation [16]. Procoagulant collagen-and-thrombin (COAT) activated platelets are characterized by high and sustained intracellular free calcium levels, loss of the mitochondrial potential, the coating of their surface by pro-hemostatic α-granule proteins, downregulation of activated GPIIb/IIIa (losing their aggregatory property), and the expression of PS to support the tenase and prothrombinase complexes for the coagulation process [14,15,16,17,18]. Investigations of the ability to generate procoagulant COAT platelets is of high clinical relevance as increased levels of procoagulant COAT platelets have been correlated with thrombotic events [19,20,21] while low levels were associated with a bleeding diathesis and its severity [22,23,24,25].

Moreover, because the procoagulant activity is a dichotomous facet of platelet activation, complementary and diverging from the aggregation endpoint [18], we were interested to characterize the functionality of procoagulant platelets in GT patients lacking platelet aggregation. Therefore, we systematically characterized platelet functions in GT patients as well as their ability to generate procoagulant COAT platelets, and we further analyzed intracellular ion fluxes upon the procoagulant response.

## 2. Results

### 2.1. Characterization of Platelet Function by Flow Cytometry

In addition to a complete medical history and traditional laboratory workup, including platelet aggregation studies, we characterized three GT patients with a comprehensive platelet phenotypic and functional analysis by flow cytometry (summarized in Table 1). The flow cytometry analysis (FCA) confirmed an absence of both components of the fibrinogen receptor, namely GPIIb (CD41) and GPIIIa (CD61), and a markedly impaired ability to bind PAC-1 following activation with increasing doses of ADP, THR or CVX. In addition, we observed a conserved platelet size and granularity, slightly in the higher range for the patient with GT number 3 (PAT_GT3), and a conserved surface density of the receptors for the von Willebrand factor (GPIb and GPIX) and the collagen receptor GPVI. The second collagen receptor GPIa (CD49b), which mainly supports platelet adhesion, was decreased in the three patients. The higher surface density measured for PAT_GT3 is corrected when data are normalized according to the platelet size (data not shown).

Moreover, we observed a conserved content of dense granule, assessed by mepacrine staining, with normal secretion following THR activation but slightly reduced after CVX for PAT_GT1 and PAT_GT3. Alpha-granule secretion following increasing doses of ADP, THR (except the highest dose), and CVX was conserved for two patients; only one patient (PAT_GT1) showed a reduced surface expression of P-selectin following stimulation with increasing concentrations of the agonists.

In our FCA, we also systematically investigated the ability to generate procoagulant COAT platelets following the combined activation with CVX-plus-THR. Interestingly, all three GT patients were revealed to generate a higher percentage of procoagulant COAT platelets or at least in the high range (for PAT_GT3) compared to normal subjects.

### 2.2. Procoagulant COAT Platelet Formation in GT Patients

Because of the consistency of our observation, we decided to investigate further the high percentage of procoagulant COAT platelets measured in the three GT patients. First, this aspect of platelet function was replicated (*n* = 2–7) among these three GT patients, and we confirmed that their individual mean value is in the high range or above our in-house reference range (Figure 1). The group of healthy donors (HD) reach a median value of 39%, with the 25th-percentile (lower bar) at 33% and the 75th-percentile (upper bar) at 44%. Taken together, the percentage of procoagulant COAT platelets generated by the three GT patients is statistically different compared to the group of HD, with PAT_GT1 reaching a value of 54% ± 2% (mean ± standard deviation). PAT_GT2 attained a value of 57% ± 10.3%, and PAT_GT3 with a value of 56% ± 5%.

### 2.3. Intracellular Ion Fluxes Following CVX-Plus-THR Platelet Activation

Second, we visualized the effects of combined activation CVX-plus-THR on ion kinetics in GT patients, especially on calcium, which is an important player of the procoagulant response (Figure 2). To assess ion mobilization, platelets were loaded with the respective ion probes (Fluo-3 for calcium, ING-2 for sodium and or IPG-2 for potassium) and fluorescence was acquired over time [26].

As illustrated in Figure 2A, the combined platelet activation by CVX-plus-THR resulted in a rapid increase in calcium mobilization. We observed a rapid and sustained calcium mobilization in platelets from HD as well as in the three GT patients. Interestingly, the initial peak mobilization and the time-dependent mobilization are constantly higher in the GT patients than in the control population.

Intracellular sodium (Figure 2B) was also characterized by a rapid mobilization after activation followed by a gradual efflux (progressive fluorescence decline) over time. This behavior was observed in HD as well as in GT patients. GT patients seemed to have a slightly higher sodium mobilization than the donors, in particular PAT_GT1 and PAT_GT2.

Cytosolic potassium (Figure 2C) showed a progressive decrease in efflux after activation, with the rate of decrease seemingly attenuating over time. In this situation, GT patients did not greatly seem to differ in their kinetics with the HD group, with the sole exception of PAT_GT3 who reached rapidly lower potassium levels following the strong CVX-plus-THR activation.

### 2.4. Intracellular Calcium during the First Three Minutes after Activation

It has been described that procoagulant COAT platelets are generated in the order of 1–2 min after combined CVX-plus-THR activation [18,27]. Since GT patients revealed the ability to generate a higher proportion of procoagulant COAT platelets with a global higher calcium mobilization, we decided to better dissect the calcium kinetics in each GT patient individually with a special focus on this early time frame. Moreover, we co-stained platelets with Annexin V to discriminate the procoagulant COAT platelets, identified as Annexin-V-positive. Therefore, for each GT patient, we individually evaluated the first 3 min post-activation (Figure 3) in comparison with kinetics from HD. In addition to the whole platelet kinetics, we are also able to appreciate the calcium mobilization in Annexin-V-positive platelet as soon as they are generated.

As already shown in the previous experiments (Figure 2A), CVX-plus-THR activation leads to a significant increase in calcium mobilization. Within the expected time frame, procoagulant COAT platelets began to appear in the HD kinetics as well as in each GT patient. Subsequently, as indicated in Figure 3A, PAT_GT1 showed a higher calcium mobilization in generated procoagulant COAT platelets than the ones in the HD pool. Figure 3B and 3C confirmed this observation for PAT_GT2 and PAT_GT3, respectively. In this context, it was remarkable that all three GT patients showed a higher initial calcium mobilization following activation in their procoagulant COAT platelet population compared to the HD group (Figure 3A–C).

### 2.5. GPIIb/IIIa Antagonists and Their Effect on COAT Platelet Formation

With a higher capacity for the formation of procoagulant COAT platelets in GT patients, the GPIIb/IIIa receptor—missing in this patient population—could be a critical regulator of intracellular calcium mobilization and procoagulant COAT platelet formation.

GPIIb/IIIa antagonists, in our case eptifibatide and tirofiban, seemed an enticing and simple approach to block the GPIIb/IIIa receptor and thus mimic the effect of GT pathophysiology. This approach would potentially allow the evaluation of GPIIb/IIIa blockage on procoagulant COAT platelet formation in an in vitro model.

Platelets from HD were pretreated with increasing concentrations of eptifibatide or tirofiban and were activated with CVX-plus-THR. Then, we evaluated the percentage of Annexin-V-positive platelets, namely procoagulant COAT platelets, at each drug concentration point.

Figure 4 displays the relative change in Annexin-V-positive events with respect to the antagonist vehicle (H_2_O or DMSO). The data did not show any remarkable and meaningful difference between basal conditions and increasing concentrations of either eptifibatide (Figure 4A) or tirofiban (Figure 4B).

## 3. Discussion

Blood platelets are crucial and active players of primary and secondary hemostasis. They contribute to forming a stable hemostatic plug in order to stop bleeding and prevent further blood loss. In order to achieve an efficient hemostatic response, activation of the fibrinogen receptor GPIIb/IIIa and to another extent surface expression of negatively charged amino-phospholipids are both essential for platelet bridging and aggregation, and clot stabilization by fibrin, respectively [12].

GT is a rare autosomal recessive inherited platelet dysfunction. The underlying deficiency is defined by the dysfunctional receptor, low expression or complete absence thereof [8]. Due to the absence or dysfunction of the GPIIb/IIIa receptor, activated platelets fail to aggregate with each other by means of fibrinogen binding [3,8].

At our institution, we systematically apply FCA as an extension to our diagnostic workup for patients with a bleeding diathesis in whom laboratory analysis could not identify a cause ([25] and Adler et al., manuscript in preparation) or in selected instances. In the case of these three GT patients, after a complete clinical and laboratory workup, including platelet aggregation tests, FCA confirmed the absence of GPIIb/IIIa at the platelet surface in all three GT patients and the impaired binding of PAC-1 following activation with increasing doses of ADP, THR, and CVX. Moreover, GT patients were revealed to have conserved platelet size and morphology, and the surface density for other GP was normal, as already documented [28,29,30]. A single exception was for GPIa. GPIa is a subunit of the adhesion collagen receptor GPIa/IIa (integrin α2β1), which plays an essential role to mediate platelet adhesion to collagen. It is known that the single nucleotide polymorphism C807T in the integrin alpha 2 gene (*ITGA2*) correlates with platelet GPIa/IIa density [31]. This gene polymorphism—with 807T allele expressing the highest receptor density, 807C the lowest and heterozygous individual expressing intermediate levels—was documented to be present in GT population, and GT patients with homozygous 807C allele were correlated with clinical bleeding severity [7,8]. The lower alpha-granule secretion observed in PAT_GT1 is congruent with its slightly lower granularity according to *SSC* properties. Common to two of our GT patients is the lower dense-granule secretion following CVX activation. Interestingly, the initial phase of GPIIb/IIIa outside-in signaling resembles that induced by the collagen receptor GPVI, with activation of Src-family kinases and Syk [3,32]. Therefore, the lacking outside-in signaling in our GT patients would impair the part of the activation pathway needed for appropriate granule secretion.

The original contribution of our work is the investigation of the procoagulant platelet activity in GT patients. Not much is known about the characteristics of procoagulant COAT platelets in the context of dysfunctional hemostasis. By confirming the presence of procoagulant COAT platelets in our GT patients and comparing them to a pool of HD, we could indeed establish a higher potency of procoagulant platelet formation in our GT patients (Figure 1).

To our knowledge, the capacity of procoagulant COAT platelet formation in GT patients has been poorly evaluated, in particular, it has not been comprehensively compared to reference ranges established in healthy subjects [25,33]. In the few publications present in the literature, the number of procoagulant platelets in GT patients differ and they have lower values than the ones we obtained [34,35,36,37,38]. However, the sample preparation in these studies (gel-filtration or washed platelets) differs from our protocol (diluted PRP). Of note, it was reported that, at least in GT patients, different platelet reactivity was observed in gel-filtered platelets compared to diluted-PRP [39] as well as according to the experimental setting (such as the use of stirring or not) [40]. Therefore, we can barely compare our results with the sparse data present in the literature and we rely on the comparison with our in-house ranges established with the same technique in HD.

The investigation of procoagulant COAT platelets in the context of GT is also of great interest in order to study and explore the role of GPIIb/IIIa in the formation of procoagulant COAT platelets. Indeed, COAT platelets gradually inactivate the GPIIb/IIIa fibrinogen receptor to become procoagulant, through surface exposure of PS. We were therefore interested to explore if we could take advantage of this natural model in order to investigate and better understand the role of GPIIb/IIIa, increasing the knowledge already generated by several studies in the past [14,17,18].

To this end, we used a flow-cytometry based approach to monitor ion fluxes (notably calcium, sodium, and potassium) in our three patients affected by GT. Moreover, flow cytometry allowed for the subsequent differentiation and selective analysis of procoagulant COAT platelets formed from non-COAT platelets [18,27].

We could characterize ion mobilization in GT patients during the procoagulant response. All our GT patients showed a higher and sustained calcium mobilization compared to the HD platelets. The mobilization of sodium replicates the known behavior in procoagulant platelets [27,41]. We also demonstrated a gradual potassium efflux over time after combined activation. This is congruent with the concept of calcium mobilization facilitating potassium efflux [26], acting on specific potassium channels on the platelet membrane [42,43,44]. Potassium efflux in our GT patients (except one) did not seem to differ from HD kinetics. As we already demonstrated in our previous publication [26], the concentration of calcium attained during the procoagulant response in HD already saturated the efflux by potassium channels (such as Gardos channels) [45]. Of note, further calcium mobilization in GT patients will not affect potassium kinetics.

An important pivot point that influences the generation of procoagulant COAT platelets is the high and sustained calcium level reached after combined activation by CVX-plus-THR [18,26,27,46,47]. Procoagulant platelets are generated in the time frame of about 1–2 min after strong activation [18,27]. For that reason, we focused our kinetics on the calcium mobilization within the first three minutes (Figure 3). Interestingly, an initial overshoot occurs right at the beginning of COAT platelet formation and stabilizes progressively on a constantly high level, which is congruent with their enhanced formation. Our results seem to suggest that a different activating or a missing inhibitory mechanism could enable GT patients to attain higher levels of calcium and generate more procoagulant COAT platelets. The missing GPIIb/IIIa receptor could be implicated in this observation. Of note, the fact that procoagulant COAT platelets inactivate the GPIIb/IIIa receptor while pro-aggregatory platelets continue to express it in its active conformation [18] strongly suggests a regulatory role of the fibrinogen receptor in the process of COAT platelet formation. It has been reported that outside-in signaling mediated by GPIIb/IIIa engagement can also inhibit additional platelet activation [48,49,50]. In particular, Rosado et al. demonstrated that occupancy of GPIIb/IIIa by fibrinogen induces inhibiting signaling which prevents cytoskeletal reorganization and limits additional store-operated calcium entry [49]. Therefore, as GPIIb/IIIa-mediated outside-in signaling serves as a regulating point not only for consolidating aggregation but also for limiting further platelet activation and calcium entry [49,50], a missing inhibitory mechanism (due to lack of the receptor) could explain a higher calcium influx in GT patients and in turn regulate the calcium influx in individuals not affected by GT.

Obviously, a limitation of this study is the small sample size of GT patients. To consolidate our data, we based our evaluation on several measurements in each patient and, in addition, we evaluated the use of GPIIb/IIIa antagonists to mimic the effect of GT pathophysiology. This would have offered a convenient and easy approach to mimic the effects of GT in a simple in vitro model. However, results seem to indicate that the inhibitory process of either eptifibatide or tirofiban does not influence the mechanism of procoagulant COAT formation in treated HD platelets. Even in the literature, a set of conflicting reports pointed out the opposite effects of GPIIb/IIIa antagonists on the procoagulant activity of platelets. Some seem to indicate that the use of GPIIb/IIIa antagonists is associated with a higher amount of procoagulant COAT platelet formation in treated platelets [51,52]—which could be congruent with the effect observed in our GT patients—while others were not able to corroborate these findings [35,36,53]. Of note, both inhibitors used here specifically interact with the ligand-binding site of GPIIb/IIIa and block fibrinogen binding and therefore platelet aggregation. However, eptifibatide and tirofiban both cause a conformational change of GPIIb/IIIa and can still induce a receptor outside-in signaling [54]. For that reason, this does not mimic the real situation of GT pathophysiology lacking this specific pathway.

To conclude, our diagnostic FCA confirmed a classic GT phenotype in these three patients. For the first time, we point out a higher ability to produce procoagulant COAT platelets, associated with higher calcium mobilization in GT. This could be a compensatory effect peculiar to GT patients alleviating their bleeding diathesis. However, while some suppressing signaling linked to GPIIb/IIIa engagement were already highlighted [49,50], it remains to be elucidated in detail how and when these potentially inhibitory or regulatory signaling processes are involved in the modulation of the procoagulant response. Further fine-tuned modulations of the outside-in signaling for example with the use of RUC-4 [55] or RGT-peptide [56,57] would possibly better mimic the GT phenotype, and this will help to bring new insights by characterizing platelet activation signaling and mechanisms leading to a procoagulant activity.

## 4. Materials and Methods

### 4.1. Material

Fluo-3 AM was acquired from Thermo Fisher Scientific (Waltham, MA, USA). ION NaTRIUM Green-2 AM and ION Potassium Green-2 AM were purchased from ION Biosciences (San Marcos, TX, USA). Convulxin was a kind gift from Prof. K.J. Clementson (Bern, Switzerland) [58]. Thrombin was purchased from Siemens (Zürich, Switzerland). Cy5 Annexin V and Annexin V Buffer containing calcium (140 mM NaCl, 2.5 mM CaCl_2_, 1 mM HEPES, pH 7.4) were from Becton Dickinson (Allschwil, Switzerland). Phycoerythrin (PE)- or fluorescein isothiocyanate (FITC)-conjugated antibodies for flow cytometric analyses were from either Becton Dickinson or Dako. Dimethyl-sulfoxide (DMSO), calcium ionophore A23187, Annexin-V-FITC FLUOS Staining Kit, eptifibatide acetate and tirofiban hydrochloride monohydrate were acquired from Merck Sigma Aldrich (St. Louis, MO, USA). To inhibit clot formation Gly-Pro-Arg-Pro-OH (GPRP H2935) from Bachem (Bubendorf, Switzerland) was used. Tyrode’s buffer was produced in our laboratory (137 mM NaCl, 2.8 mM KCl, 12 mM NaHCO_3_, 1 mM MgCl_2_, 0.4 mM NaH_2_PO_4_, 10 mM HEPES, 3.5 mg/mL bovine serum albumin, 5.5 mM glucose, adjusted to pH 7.4 with NaOH).

### 4.2. Healthy Donors and Patients with Glanzmann Thrombasthenia

Healthy donors were recruited from the local blood donation service or within our division (age 34 ± 12 [mean ± standard deviation]; range min–max: 20–65; female 63%). Donors did not ingest any medication influencing platelet function during the previous 10 days, and they gave written informed consent for the research use of their blood samples.

We enrolled the three adult GT patients (two males and one female, age 19, 28, and 51 years old) followed at our outpatient clinic of the Lausanne University Hospital in the Service of Hematology. GT diagnosis was based on clinical manifestations, platelet aggregation analysis, and platelet characterization by flow cytometry analysis that confirmed complete absence of GPIIb/IIIa.

The study was conducted according to the rules of our institution, the local independent ethics committee (CER-VD project n. PB_2018-00205 (422/14); 3rd October 2018), and the Declaration of Helsinki.

### 4.3. Blood Collection and Preparation of Platelet Rich Plasma

A 17 gauche needle was used to collect whole blood from the antecubital vein into S-Monovettes^®^ (Sarstedt AG and Co. KG, Nümbrecht, Germany) containing 0.129 mol/L of buffered trisodium citrate (pH 5.5). Platelet-rich-plasma (PRP) was prepared by centrifugation at 200 g for 10 min at 22 °C and the subsequent platelet count normalized to 300 × 10^9^ /L in Tyrode’s buffer.

### 4.4. Flow Cytometry Analysis

Flow cytometry analysis was performed in our diagnostic laboratory as published ([25] and Adler et al., manuscript in preparation), and analyzed with a BD Accuri C6 flow cytometer (Becton Dickinson, Allschwil, Switzerland).

Surface specific antigens were analyzed with fluorescence-conjugated monoclonal antibodies (mAbs). Basal (non-stimulated) platelets were incubated for 15 min in the dark at room temperature (RT) with mAbs directed against human CD41 (GPIIb), CD61 (GPIIIa), CD42b (GPIbα), CD62P (P-selectin), CD42b (GPIX), CD49b (GPIa), and GPVI.

Activation of the fibrinogen receptor GPIIb/IIIa and alpha-granule secretion were assessed with binding of fluorescence-conjugated anti-human PAC-1 (anti-CD41/CD61 complex) or anti-human CD62P mAbs, respectively. Platelets were stimulated with increasing concentrations of ADP (0.5, 5.0, and 50 μM), convulxin (5, 50, and 500 ng/mL), and thrombin (0.005, 0.05, and 0.5 U/mL) for 10 min in the dark at 37 °C in presence of PAC-1-FITC and anti-CD62P-PE.

Dense granule content and secretion were evaluated with mepacrine stain, a dye that is rapidly uptaken and localized in platelet dense granules. Platelets were loaded for 30 min at 37 °C with two concentrations of mepacrine, namely 0.17 μM and 1.7 μM. Secretion of dense granules was measured after stimulation of mepacrine-stained platelets with thrombin (0.5 U/mL) or CVX (500 ng/mL) for 10 additional minutes at 37 °C. The secretion of dense granules was calculated as the decrease in fluorescence in stimulated platelets relative to the basal fluorescence (mepacrine uptake) [25].

### 4.5. Procoagulant COAT Platelet Generation

Procoagulant COAT platelets were induced by combined stimulation with CVX and THR as already described [16,18]. Briefly, platelets (5 × 10^9^ /L) were stimulated in calcium buffer (from FLUOS Staining Kit) with CVX (500 ng/mL) and THR (0.5 U/mL) for 8 min at 37 °C, and expression of externalized PS was detected with Annexin V-FITC (FLUOS Staining Kit) binding by flow cytometry (BD Accuri C6). For experiments using GPIIb/IIIa antagonists, platelets were pretreated with increasing doses of eptifibatide (stock solution 2 mM in distillate H_2_O MQ) or tirofiban (stock solution 10 mM in DMSO) and activated as described above.

### 4.6. Kinetic Ion Flux Measurements

Cell-permeant selective fluorescent ion indicators were used to measure intracellular ion fluxes. Calcium was detected with Fluo-3 AM (λ_em_ 526 nm, λ_ex_ 506 nm), sodium with ION NaTRIUM Green-2 (ING-2) AM (λ_em_ 540 nm, λ_ex_ 488–517 nm) and potassium with ION Potassium Green-2 (IPG-2) AM (λ_em_ 540 nm, λ_ex_ 488–517 nm).

Further detailed method is described in our previous publications [26,27]. In summary, PRP was adjusted to 30 × 10^9^/L in Tyrode’s buffer and was incubated in the dark with either Fluo-3 AM (final concentration 2 μM), ING-2 AM (7.5 μM) or IPG-2 AM (0.1 μM) at RT for 60 min. The loaded platelets were diluted 1:5 (final 5 × 10^9^/L) with a calcium-containing buffer (final 2 mM). Moreover, to delineate the distinct kinetic behaviors in platelet subpopulations, i.e., procoagulant COAT platelets, Annexin V Cy5 (λ_em_ 670 nm, λ_ex_ 625–650 nm) was added to the sample in a 1:20 ratio.

Events were acquired with slow speed on BD Accuri C6 flow cytometer. Platelets were gated (P1) according to their *FSC/SSC* properties. For ion kinetics, we set acquisition parameters on *Time* for the abscissa and *FL1-A* (488:530/30:[Fluo-3; ING-2; IPG-2]) for the ordinate. After the acquisition of a stable baseline fluorescence of 4 min, the process was halted and platelet activation was induced by the combined addition of CVX (500 ng/mL) and THR (0.5 U/mL). Acquisition was resumed and continued for 10 min.

### 4.7. Data Analysis

For kinetics data, Median Fluorescence Intensities (MFI) from raw data were extracted with the Kinetics module analysis from FlowJo™ Software (for Windows) Version 10.2 (Ashland, OR: Becton, Dickinson and Company, Franklin Lakes, NJ, USA; 2019) as described in [26]. As non-ratiometric dye fluorescence is related to targeted ion binding and consequently variations of the dye fluorescence reflect changes in ion concentrations, fluorescence of ion probes is presented in fold changes. The reference value was set as the mean fluorescence of the baseline (4-min measurement). Figures were generated using GraphPad Prism version 8.0.1 for Windows (GraphPad Software, San Diego, CA, USA). Data are displayed as mean ± standard error of mean (SEM) from separate experiments with platelets from different donors.

Statistical analyses were performed in GraphPad Prism. Nonparametric tests were used for comparisons. Patient data were compared to controls using the Mann–Whitney test. A *p* < 0.05 was considered statistically significant.

## Figures and Tables

**Figure 1 ijms-21-09515-f001:**
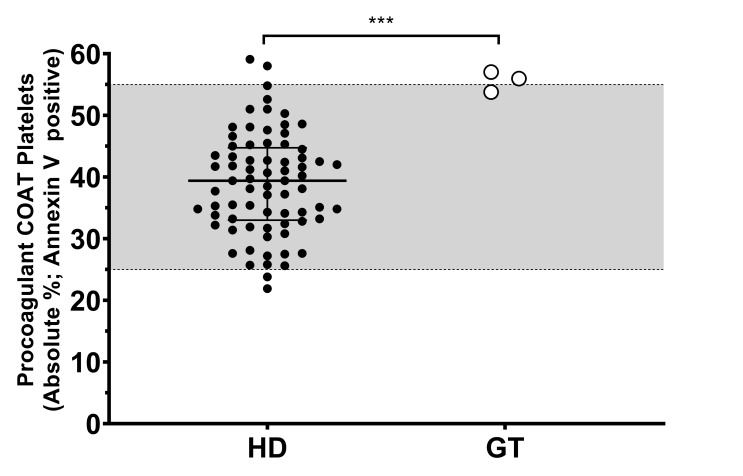
Procoagulant collagen-and-thrombin (COAT) platelet formation in patients with Glanzmann thrombasthenia and healthy donors. Absolute percentage of the Annexin-V-positive platelets following activation with convulxin-plus-thrombin in healthy donors (HD) and means of at least two replicates for patients with Glanzmann thrombasthenia (GT). The grey zone (25–55%) indicates the in-house reference ranges (2.5–97.5 percentiles). Statistical significance was determined by Mann–Whitney test, *** *p* < 0.001.

**Figure 2 ijms-21-09515-f002:**
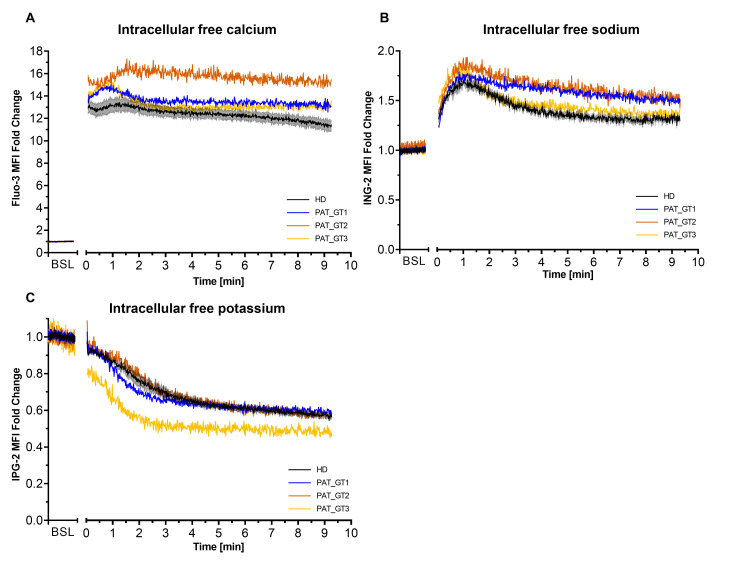
Intracellular ion kinetics following platelet activation with convulxin and thrombin. Platelets from healthy donors (HD) or patients with Glanzmann thrombasthenia (GT) were loaded with ion fluorescent indicator. Fluorescence from ion probes was monitored in FL1 (488:530/30) and median fluorescence intensity (MFI) fold change (relative to the baseline [BSL]) for the whole platelet population was plotted over time. (**A**) Calcium monitoring with Fluo-3. (**B**) Sodium monitoring with ION NaTRIUM Green-2 (ING-2). (**C**) Potassium monitoring with ION Potassium Green-2 (IPG-2). Data from HD represented as mean ± SEM, from experiments with platelets from different donors (*n* = 6). Data for each GT patient (named PAT_GT1, PAT_GT2, and PAT_GT3) are a mean of individual replicates, *n* = 3 for PAT_GT1, *n* = 1 for PAT_GT2, and *n* = 2 PAT_GT3.

**Figure 3 ijms-21-09515-f003:**
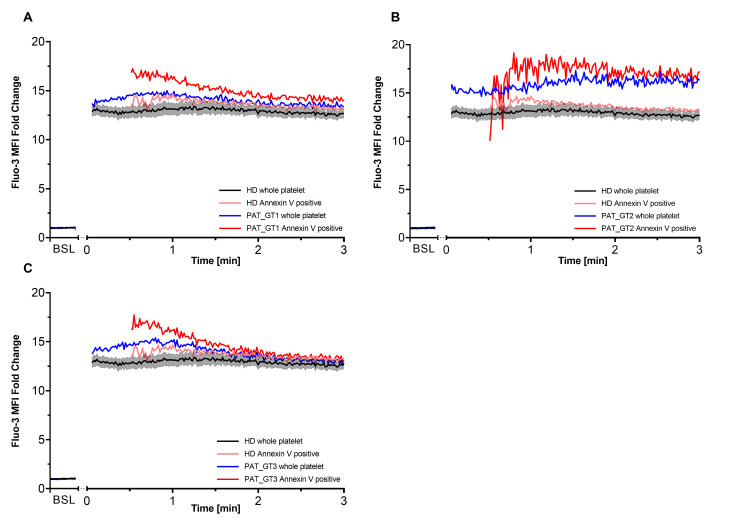
Intracellular calcium within first 3 min after activation. Each of the three patients with Glanzmann thrombasthenia (GT) had their first 3 min kinetics after activation plotted with respect to their Fluo-3 median fluorescence intensity (MFI) fold change compared to healthy donors (HD). Co-staining with Cy5 Annexin V allowed differentiating procoagulant COAT platelets (Annexin V positive) from the whole platelet population. (**A**) PAT_GT1; (**B**) PAT_GT2; (**C**) PAT_GT3. HD data are mean ± SEM from experiments with platelets from different donors (*n* = 6). Data for each GT patient are a mean of individual replicates, *n* = 3 for PAT_GT1, *n* = 1 for PAT_GT2, and *n* = 2 PAT_GT3. Baseline, BSL.

**Figure 4 ijms-21-09515-f004:**
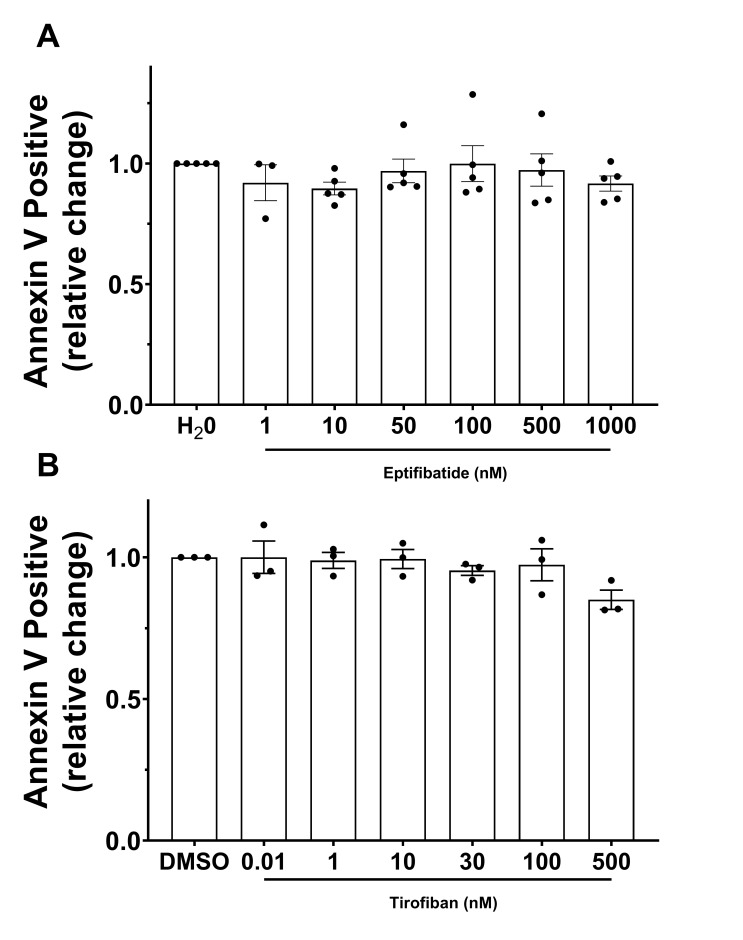
GPIIb/IIIa-antagonists effect on the formation of procoagulant COAT platelets. Platelets were pretreated with increasing doses of (**A**) eptifibatide or (**B**) tirofiban, and activated with convulxin-plus-thrombin. Relative change of the fraction of Annexin-V-positive platelets compared to control condition (vehicle H_2_O or DMSO, respectively) is represented as mean ± SEM from experiments with platelets from different donors (*n* = 3–5).

**Table 1 ijms-21-09515-t001:** Patient Characteristics Revealed by Flow Cytometric Analysis and In-House Reference Ranges.

Characterization	Indicator	Unit	PAT_GT1	PAT_GT2	PAT_GT3	In-House Ranges (*n* = 73; 2.5–97.5 Percentiles)		
**Size**	FSC	MFI	74′739	93′372	125′674	74′002	-	126′189
**Granularity**	SSC	MFI	**6′322 ***	7′588	9′610	7′073	-	10′808
**Surface GP Markers**								
GPIIb	anti-CD41 mAb	MFI	**184 ***	**445 ***	**225 ***	14′392	-	21′923
GPIIIa	anti-CD61 mAb	MFI	**229 ***	**640 ***	**328***	25′111	-	39′696
GPIb	anti-CD42b mAb	MFI	23′539.0	20′565.0	**31′463 ***	18′647	-	28′661
GPIX	anti-CD42a mAb	MFI	27′033	22′486	**32′646 ***	19′407	-	27′401
GPVI	anti-GPVI mAb	MFI	4′738	5′491	**7′851 ***	4′584	-	7′518
GPIa	anti-CD49b mAb	MFI	**1′244 ***	**1′291 ***	**1′463 ***	1′485	-	4′227
**Dense granules**								
Content	Mepacrine 0.17 μM	MFI	374	404	398	278	-	502
After CVX (500 ng/mL)		MFI	322	304	350	215	-	377
CVX-induced secretion		%§	**−13.9 ***	−25	**−12 ***	−14	-	−41
After THR (0.5 U/mL)		MFI	170	176	180	162	-	188
THR-induced secretion		%§	−55	−56	−55	−37	-	−64
Content	Mepacrine 1.7 μM	MFI	618	632	701	441	-	817
After CVX (500 ng/mL)		MFI	563	516	651	294	-	565
CVX-induced secretion		%§	**−9 ***	−18	**−7 ***	−16	-	−53
After THR (0.5 U/mL)		MFI	199	200	216	181	-	229
THR-induced secretion		%§	−68	−68	−69	−58	-	−74
**Alpha granules**	anti-CD62P mAb							
Baseline		Absolute %	0.2	0.8	1.1	0.2	-	4.9
		MFI	182	194	183	168	-	285
ADP 0.5 μM		MFI	**265 ***	527	314	297	-	958
ADP 5 μm		MFI	**524 ***	1′374	704	620	-	2′564
ADP 50 μm		MFI	**529 ***	1′322	731	766	-	3′081
THR 0.005 U/mL		MFI	376	1′979	1′510	250	-	2′735
THR 0.05 U/mL		MFI	**4′304 ***	6′067	6′221	5′062	-	9′957
THR 0.5 U/mL		MFI	**5′145 ***	**6′585 ***	**6′719 ***	6′757	-	11′321
CVX 5 ng/mL		MFI	1′682	4′362	1′908	789	-	9′312
CVX 50 ng/mL		MFI	**3′614 ***	5′918	5′781	4′643	-	10′030
CVX 500 ng/mL		MFI	**4′132 ***	6′058	6′132	5′609	-	10′891
**GPIIb/IIIa Activation**	anti-CD41/CD61 (PAC−1) mAb							
Baseline		MFI	**150 ***	**376***	**153 ***	469	-	1′145
ADP 0.5 μM		MFI	**147 ***	**583 ***	**151 ***	1′564	-	5′049
ADP 5 μm		MFI	**155 ***	**821 ***	**144 ***	4′356	-	12′182
ADP 50 μm		MFI	**151 ***	**776 ***	**145 ***	6′323	-	16′678
THR 0.005 U/mL		MFI	**151 ***	**1′128 ***	**141 ***	980	-	5′116
THR 0.05 U/mL		MFI	158 *	**1′886 ***	**105 ***	9′060	-	22′867
THR 0.5 U/mL		MFI	**160 ***	**1′621 ***	**103 ***	15′017	-	28′176
CVX 5 ng/mL		MFI	**152 ***	**1′656 ***	**128 ***	2′026	-	12′135
CVX 50 ng/mL		MFI	**150 ***	**1′544 ***	**101 ***	5′605	-	14′931
CVX 500 ng/mL		MFI	**144 ***	**1′574 ***	**v96 ***	6′293	-	14′789
**Procoagulant Activity**	Annexin V							
Baseline		Absolute %	4.1	5.9	1.6	0.8	-	4.0
Ionophore		Absolute %	99	99	99	96	-	100
COAT platelets (CVX + THR)		Absolute %	**57 ***	**64 ***	50	25	-	55
		MFI	23′561	23′403	31′741	13′645	-	116′434

Abbreviations: ADP, adenosine diphosphate; CVX, convulxin; FSC, forward scatter; mAb, monoclonal antibody; MFI, median fluorescence intensity; SSC, side scatter; THR, thrombin; PAT_GT#, Patient with Glanzmann thrombasthenia number #. ^§^ Percentage decrease in fluorescence in stimulated platelets relative to the basal fluorescence (mepacrine uptake). * values out of the reference range.

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
