# Peer review of "Characterization of Procoagulant COAT Platelets in Patients with Glanzmann Thrombasthenia"

_ijms, 2020, doi:10.3390/ijms21249515_

Round 1

Reviewer 1 Report

Aliotta and colleagues made a word on the subpopulation of platelets displaying procoagulant properties (opposite to aggregating platelets) in Glanzmann thrombasthenia (GT). They compared 3 GT patients with healthy donors (HD) and found that despite the limited activation of platelets toward an aggregating phenotype they were able to become procoagulant and mobilized more cytosolic calcium.

Major comment: These findings are new and focus on an interesting topic that may help to stratify patients and better evaluate the risk of bleeding (and it also applies to other diseases related to platelet activity). As the number of patients is limited the authors should present all patients and HD characteristics. Are the cytometry results similar if only age and sex match HD are selected (it is not necessary to present this in the manuscript)? In figure 2 the number of HD used is not specified. Figure 4 should be presented as bars with dot plots.

Minor comment:

In the abstract “which acts a fibrinogen receptor” may be “which acts as a fibrinogen receptor”

The definition of COAT is lacking in the abstract.

Reviewer 2 Report

In this paper, the authors show that the proportion of proocoagulant platelets is greater in patients with Glanzmann's Thrombasthenia, compared to people without a platelet disorder.   The link to a higher cytosolic Ca2+ signal is interesting and may help us better understand the pathophysiology of this and other inherited platelet disorders. There are a few minor amendments I would like to see prior to publication of this paper.

  1. Was there a difference in the baseline fluorescence of the GT and control groups?  As the data is normalised to the baseline fluorescence through an F/F0 calculation, it is important to report these values to ensure that normalisation is fair.
  2. Previous studies have shown that fibrinogen binding inhibits store-operated Ca2+ entry in human platelets (Josado et al (2001) Blood 97: 2648-2656).  This work could explain the differences observed in this paper, and so should be included in the conclusion.  Interestingly this group found that blocking fibrinogen binding with RGDS peptide was able to inhibit this effect.  Can the authors place these results in the context of their own findings?
  3. There is a correlation in the findings of a higher baseline pro-coagulant activity seen in PAT_GT1 and PAT_GT2 levels, and higher post-stimulation levels compared to PAT_GT3 platelets (which is just within the normal range).  Can your data be explained by pre-sensitized platelets in these samples leading to higher cytosolic Ca2+ signals and thus greater pro-coagulant responses?
  4. I presume that all donors gave written informed consent?  It would be good to see that explicitly stated in the materials and methods

Round 2

Reviewer 1 Report

The authors made a good review and answered all my remarks and I have no further questions.